# Bortezomib in Combination with Physachenolide C Reduces the Tumorigenic Properties of KRAS^mut^/P53^mut^ Lung Cancer Cells by Inhibiting c-FLIP

**DOI:** 10.3390/cancers16030670

**Published:** 2024-02-04

**Authors:** Thanigaivelan Kanagasabai, Zerick Dunbar, Salvador González Ochoa, Tonie Farris, Sivanesan Dhandayuthapani, E. M. Kithsiri Wijeratne, A. A. Leslie Gunatilaka, Anil Shanker

**Affiliations:** 1Department of Biomedical Sciences, School of Graduate Studies, Meharry Medical College, Nashville, TN 37208, USA; tkanagasabai@mmc.edu (T.K.); tfarris20@email.mmc.edu (T.F.); 2Department of Microbiology, Immunology & Physiology, School of Medicine, Meharry Medical College, Nashville, TN 37208, USA; zdunbar18@email.mmc.edu; 3Department of Biochemistry, Cancer Biology, Neuroscience and Pharmacology, School of Medicine, Meharry Medical College, Nashville, TN 37208, USA; sgonzalezochoa@mmc.edu; 4Central Research Facility, Santosh Deemed to Be University, Ghaziabad 201009, UP, India; snesan397@gmail.com; 5Southwest Center for Natural Products Research, School of Natural Resources and the Environment, College of Agriculture, Life and Environmental Sciences, The University of Arizona, Tucson, AZ 85719, USA; kithsiri@cals.arizona.edu (E.M.K.W.);

**Keywords:** lung cancer, natural product, withanolides, physachenolide C, bortezomib, c-FLIP, KRAS, P53

## Abstract

**Simple Summary:**

c-FLIP is a master anti-apoptotic regulator associated with resistance to various therapeutic options in different cancer types. Thus, exploring the combinatorial antitumor effects of a natural product, physachenolide C (PCC) and bortezomib, could lead to an effective, non-toxic therapeutic regimen against KRAS^mut^/P53^mut^ lung cancer cells. Our results showed that the bortezomib–PCC combination was more effective in reducing the viability and inhibiting migration and invasion of cancer cells. Additionally, c-FLIP protein expression was significantly inhibited along with a substantial reduction in the critical parameters of cellular metabolism in cancer cells. Strikingly, tumor growth inhibition was much more effective in tumor xenograft mice models with an enhanced efficacy without causing significant side effects.

**Abstract:**

Background: Defects in apoptosis regulation are one of the classical features of cancer cells, often associated with more aggressiveness and failure to therapeutic options. We investigated the combinatorial antitumor effects of a natural product, physachenolide C (PCC) and bortezomib, in KRAS^mut^/P53^mut^ lung cancer cells and xenograft mice models. Methods: The in vitro anticancer effects of the bortezomib and PCC combination were investigated using cell viability, migration, and invasion assays in 344SQ, H23, and H358 cell lines. Furthermore, the effects of combination treatment on the critical parameters of cellular metabolism, including extracellular acidification rate (ECAR) and mitochondrial oxidative phosphorylation based on the oxygen consumption rate of cancer cells were assessed using Seahorse assay. Finally, the antitumor effect of the bortezomib (1 mg/kg) and PCC (10 mg/kg) combination was evaluated using xenograft mice models. Results: Our data showed that the bortezomib–PCC combination was more effective in reducing the viability of lung cancer cells in comparison with the individual treatments. Similarly, the combination treatment showed a significant inhibition of cell migration and invasion of cancer cells. Additionally, the key anti-apoptotic protein c-FLIP was significantly inhibited along with a substantial reduction in the key parameters of cellular metabolism in cancer cells. Notably, the bortezomib or PCC inhibited the tumor growth compared to the control group, the tumor growth inhibition was much more effective when bortezomib was combined with PCC in tumor xenograft mice models. Conclusion: These findings demonstrate that PCC sensitizes cancer cells to bortezomib, potentially improving the antitumor effects against KRAS^mut^/P53^mut^ lung cancer cells, with an enhanced efficacy of combination treatments without causing significant side effects.

## 1. Introduction

Cancer is the second leading cause of death in the United States. Particularly, solid tumors are among the most common lethal cancers with poor prognosis at the metastatic stage [1]. Although checkpoint immunotherapies have recently shown some success, there is still an unmet clinical need for treating advanced solid tumors [2]. Current major treatments for cancer management are surgery, cytotoxic chemotherapy, targeted therapy, radiation therapy, endocrine therapy, and immunotherapy. Despite the various therapeutic options available, cancer cells still resist classical chemotherapeutic agents [3]. Resistance to anticancer drugs arises from a wide variety of factors and various other cellular and molecular mechanisms, including epigenetic changes, conserved by upregulated drug efflux, and genetic mutations [3]. Even though mutation in the *TP53* gene is common in almost all types of cancers [4], it is reported in more than 50% of all NSCLCs [5]. Similarly, a mutation in *KRAS* is present in about 30% of all NSCLCs [6].

Natural products are an important source of anticancer drugs, that include well-known chemotherapeutic agents such as paclitaxel, vincristine, adriamycin, and vinblastine [7,8]. Withanolides are a group of naturally occurring steroidal lactones isolated from members of the Solanaceae plant family [9]. Recent evidence has been showing that this class of natural products (NP) has shown promising anticancer properties against multiple tumor types [10,11,12,13,14,15,16,17]. Physachenolide C (PCC) is a withanolide originally obtained by epoxidation of physachenolide D isolated from *Physalis crassifolia* [16], but was later found to be a minor natural product in aeroponically cultivated plants [15]. It is a potent and selective anticancer withanolide compared to withaferin A and withanolide E [15] by sensitizing various human melanoma and renal carcinoma cells to undergo apoptosis [13].

Bortezomib, a proteasome inhibitor, is a drug approved by the U.S. Food and Drug Administration to treat multiple myeloma [18,19] and mantle cell lymphoma [20,21]. It acts by reversibly binding to the chymotrypsin-like subunit of the 26S proteasome, resulting in its inhibition and preventing the degradation of various pro-apoptotic factors [18]. Several studies demonstrated that bortezomib could sensitize mouse and human solid tumor cells to undergo apoptosis by upregulating caspase-8 activity in the death-inducing signaling complex (DISC) following death receptor ligation on tumor cells [22,23]. However, the antitumor effects of bortezomib on KRAS^mut^/p53^mut^ lung cancer cells are understudied.

Cellular FLICE (FADD-like IL-1β-converting enzyme)-inhibitory protein, also known as c-FLIP, is a master anti-apoptotic regulator and resistance factor that suppresses tumor necrosis factor-α (TNF-α), Fas-L, and TNF-α-related apoptosis-inducing ligand (TRAIL)-induced apoptosis, as well as apoptosis triggered by chemotherapy agents in various types of cancer [24]. C-FLIP is also considered the main causal factor for immune escape [25]. Increased expression of c-FLIP isoforms has been reported in various types of cancers, including colorectal [24,26,27], bladder urothelial cancer [28], cervical cancer [29], Burkitt’s lymphoma [1], non-Hodgkin’s lymphoma [30], head and neck squamous cell carcinoma (HNSCC) [31], and hepatocellular carcinoma [32]. In addition, c-FLIP upregulation is also seen in gastric cancer and plays a crucial role in lymph node metastasis [33]. It is also expressed in pancreatic and prostate cancer tissues, and the maximal c-FLIP expression was detected in castrate-resistant prostate cancer (CRPC) [34]. Therefore, targeting c-FLIP would be an ideal strategy; however, targeting c-FLIP function with small molecule ligands would be difficult since c-FLIP has significant structural similarity to caspase-8 [35]. Thus, an attempt has been made to assess whether PCC’s ability to promote apoptosis of solid tumor cells by inhibiting c-FLIP in combination with bortezomib treatment could lead to a clinically effective, non-toxic therapeutic regimen that can be used for treating metastatic solid tumors that are resistant to current conventional therapy.

## 2. Materials and Methods

### 2.1. Cell Lines and Reagents

Lung cancer cell lines NCI-H23 [H23]/ATCC^®^ CRL-5800^™^, NCI-H358 [H358]/ATCC^®^ CRL-5807 (human non-small-cell lung adenocarcinoma cell lines) were kindly provided by Dr. David P Carbone’s lab at The Ohio State University. Murine lung cancer cell line 344SQ (subcutaneous metastasis model) was kindly provided by Dr. Don L. Gibbons’s lab at The University of Texas MD Anderson Cancer Center (Houston, TX, USA). C-FLIP overexpressed R331 cell line (clone established from Renca cell line) was kindly provided by Dr. Thomas J. Sayers’s lab at the Frederick National Laboratory for Cancer Research, Frederick, MD, USA. All the cell lines were tested for mycoplasma contamination frequently before any experiments were performed. Cells were grown in RPMI-1640 medium supplemented with 10% fetal bovine serum (FBS), 1% L-glutamine, 1.5 g/L of sodium bicarbonate and 1% antibiotic–antimycotic solution (Corning, Manassas, VA, USA). The cells were carefully maintained in a humidified air/CO_2_ (19:1) atmosphere at 37 °C following standard cell culture protocol. The cells were subcultured into a new culture flask, growing to approximately 80% confluence. Only single-cell suspensions with >95% viability were injected into the experimental mice. We used antibodies against c-FLIP (#3094, 1:1000) and β-actin (#9661, 1:5000). The ECL substrate (#32132, Thermo Fisher Scientific, Waltham, MA, USA) was used for protein visualization. Sunitinib was purchased from Sigma-Aldrich Co. (St. Louis, MO, USA). All other chemicals used in this experiment were of research grade. The natural product, PCC, was obtained by epoxidation of physachenolide D isolated from *Physalis crassifolia* as described previously [15].

### 2.2. In Vitro Cell Viability

The cell viability was assessed using the crystal violet assay (Sigma-Aldrich Co., St. Louis, MO, USA). Briefly, 2 × 10^4^ cells per well were seeded into a 24-well plate and incubated overnight. Following that, cells were treated with various concentrations of bortezomib (0–100 nM) and PCC (0–2000 nM), and the plates were incubated for 48 h. After incubation, cells were rinsed with PBS and fixed with 10% formalin for 15 min. After washing, the cells were stained with 0.1% crystal violet and incubated for 30 min at room temperature. The excess stain was removed by washing the plates twice and air-drying. Then, the cell-bound crystal violet was suspended in 10% acetic acid, and the absorbance (A) was read at 570 nm using a BioTek Synergy H1 multimode plate reader (BioTek Instruments, Inc., Winooski, VE, USA).

### 2.3. MTS Cytotoxicity 

The cytotoxicity effects of bortezomib and PCC on naïve and activated CD8^+^ T cells were performed using MTS assay. Briefly, 2 × 10^5^ cells were seeded and activated with 1 μg/mL of anti-CD3 and 1 μg/mL of anti-CD28 (Biolegend, San Diego, CA, USA) with indicated treatment conditions for 24–48 h. Viable cell number was determined by the addition of CellTiter 96 Aqueous cell proliferation assay solution (MTS), plates were incubated for 2 h, and then the absorbance was measured (A) at 490 nm. Cytotoxicity was calculated as in the following formula: % cytotoxicity = [(Amedium − Atreatment)/Amedium)] × 100.

### 2.4. In Vitro Cell Migration 

The effect of bortezomib and PCC on cell migration was assessed using both Transwell and wound-healing scratch assays. Cell migration assay was performed using 6.5-mm Transwell inserts with an 8 µm polycarbonate membrane (Corning Inc., Manassas, VA, USA), following the manufacturer’s protocol. Briefly, cells were suspended in a serum-free medium with the desired treatments (344SQ-BZB 5 nM/PCC 250 nM; H23 and H358-BZB 20 nM/PCC-500 nM) added onto the top chambers, and the medium containing 10% FBS was placed in the lower chambers as a chemoattractant stimulus. Cells were incubated for 48–72 h to migrate through the porous membrane. Following incubation, the cells were washed in PBS and fixed in 4% paraformaldehyde and stained with 0.1% crystal violet solution. Following washes with water, the images were captured using a microscope (Keyence microscope, Keyence Co., Itasca, IL, USA), and the number of migratory cells were analyzed using ImageJ software (Version 1.53e). For the scratch wound assay, a confluent monolayer of cells was grown in 24-well plates, and scratches were made with 200 µL microtips followed by treating the cells with the desired concentration of bortezomib, PCC, and bortezomib and PCC combination. Images were acquired after 48 h (Keyence microscope, Keyence Co., Itasca, IL, USA), and the cell migratory effect was calculated as the percentage of the cell migration from four random microscopic fields at 20× magnification. Images were analyzed using ImageJ software (Version 1.53e).

### 2.5. Cell Invasion

For cell invasion assay, Matrigel was used in the Corning chamber. After 48–72 h, noninvasive cells on the inner membrane were gently removed by wiping with a cotton swab. The invaded cells at the bottom surface of the membrane were fixed with 4% paraformaldehyde, followed by staining with 0.1% crystal violet. Cells were counted in four random microscopic fields at 20× magnification (Keyence microscope, Itasca, IL, USA).

### 2.6. Western Blotting Analysis and Protein Stability 

Cell lysates were prepared from the above-indicated treatment groups using RIPA lysis buffer (1× PBS, 1% Nonidet P-40/Triton X-100, 0.5% sodium deoxycholate, 2 mM of EDTA, 0.1% SDS, and protease inhibitor cocktail, Roche, Basel, Switzerland). Equal concentrations of protein samples were separated by polyacrylamide gel electrophoresis. After immunoblotting, the membranes were blocked with 5% non-fat dry milk solution for 1 h. The membranes were incubated with Rabbit anti-c-FLIP antibody (c-FLIP 1:1000) for overnight incubation at 4 °C. Then, the membranes were incubated with corresponding HRP-conjugated secondary antibody, and the protein bands were visualized using the ECL chemiluminescence substrate (Amersham, Chicago, IL, USA). Β-Actin (Sigma, Saint Louis, MO, USA) band intensity was used to normalize the protein expression. The protein band intensity was quantified using ImageJ software (Version 1.53e) quantified (NIH Image, Bethesda, MD, USA). Protein stability and half-life were determined by exposing the cells to 100 μg/mL of cycloheximide (CHX, Sigma, Saint Louis, MO, USA) and collecting lysates at the indicated time points for immunoblot blot analysis described above. 

### 2.7. Metabolic Stress

H23 and H358 lung cancer cells were seeded at a density of 2 × 10^4^ cells in a Seahorse Xfe96-well plate and allowed to attach overnight before treating the cells with bortezomib and PCC. For CD8^+^ T cells, 2.5 × 10^5^ cells/well were seeded in the 96-well plate coated with Poly-d-lysine hydrobromide (0.1 g/L). CD3 and CD28 antibodies were used to activate CD8^+^ T cells for 48 h and further treating the cells with bortezomib and PCC for 24–48 h. After the treatment, the Oxygen Consumption Rate (OCR) and Extracellular Acidification Rate (ECAR) were measured using the Seahorse Xfe96 Analyzer (Agilent Technologies, Santa Clara, CA, USA) following the manufacturer’s instruction. On the day of analysis, culture medium was replaced with Seahorse XF base medium (Cat# 102353-100) supplemented with 1 mM of sodium pyruvate, 2 mM of L-glutamine and 25 mM of glucose for mitostress test, and for glycolytic assay, only L-glutamine was added to the base medium. Cells were incubated at 37 °C in a CO_2_-free incubator for 1 h. The Oxygen Consumption Rate (OCR) and Extracellular Acidification Rate (ECAR) were measured over 100 min with compounds being injected every three cycles. For the mitochondrial respiration assays, the following compounds were injected sequentially (final concentrations in the wells): oligomycin (3 μM), FCCP (0.25 μM), rotenone (1 μM), and antimycin A (1 μM) (MedChem Express LLC, Monmouth Junction, NJ, USA). For the glycolysis assays, the following compounds (Sigma-Aldrich, St. Louis, USA) were injected sequentially (final concentrations in the wells): glucose (10 mM), oligomycin (3 μM), FCCP (0.25 μM), 2-deoxy-d-glucose (100 mM). Data were normalized to the cell number using the Sulforhodamine B assay following the manufacturer’s instructions (Abcam, Waltham, MA, USA). Data analyses were performed using Wave 2.6.1 software and the Seahorse XF Cell Mito Stress Test Report Generator (Agilent Technologies, Santa Clara, CA, USA).

### 2.8. CD8^+^ T Cell Purification and Activation

Single-cell suspensions were prepared from the spleen and lymph node of Balb/c WT mice by smashing the tissue in the cell strainer. Then, CD8^+^ T cells were purified using the CD8^+^ T cell isolation kit (Miltenyi Biotec. Inc., Gaithersburg, MD, USA) following the manufacturer’s protocol. The purified CD8^+^ T cells were stimulated with 1 μg/mL of anti-CD3 and 1 μg/mL of anti-CD28 (Biolegend, San Diego, CA, USA) for 24–48 h with indicated treatment conditions or left untreated. Following treatment, the cell viability was assessed using MTS assay.

### 2.9. Xenograft Lung Tumor Model

Male and Female athymic nude mice, 8–10 weeks old (nu/nu NCr), were inoculated with 2 × 10^6^ H358 human and 344sQ murine lung cancer cell suspension in a 0.1 mL volume of Matrigel/PBS mix into the flank tissues. When the tumor reaches about ~100 mm^3^, the tumor-bearing mice were randomly divided into 4 groups: group I was the vehicle control; group II was treated with bortezomib (1 mg/kg in 10% DMSO, 40% PEG300, 5% tween-80, and 45% saline; 3 times/week; i.p. injection); group III was treated with PCC (10 mg/kg in 30% Trappsol^®^ and 30% DMSO, 3 times/week; i.p. injection); and group IV was treated with the combination of bortezomib (1 mg/kg; 3 times/week; i.p. injection) and PCC (10 mg/kg, 3 times/week; i.p. injection) for 4 weeks. Trappsol^®^ was prepared at a concentration of 43% in saline and added to PCC dropwise to yield a final concentration of 30%). The tumor volume (V) was calculated using the formula V = 1/2 × (L × W^2^), where L is the length and W is the width. All animal care and experiments were performed in accordance with the guidelines and approval of the Institutional Animal Care and Use Committee (Animal protocol approval #16-07-582 dated 23 August 2022). 

### 2.10. Chronic Toxicity Assessment In Vivo

To test the potential toxicity of bortezomib and PCC, 8- to 10-week-old in-house-bred BALB/c mice (weight, 22–30 g) were used. The experimental animals were divided into 4 groups (6 in each group). The first group was the vehicle control that was not treated with any drug; the second group was injected with bortezomib (1 mg/kg body weight, i.p. in 10% DMSO, 40% PEG300, 5% tween-80, and 45% saline); the third group was injected with PCC (10 mg/kg body weight; in 30% Trappsol^®^ and 30% DMSO, 3 times/week) and the fourth group was injected with bortezomib (1 mg/kg body weight) and PCC (10 mg/kg body weight). The injections were given intraperitoneally (i.p.) once every 2 days for the duration of 4 weeks. All the animals were monitored twice a day for harmful side effects such as allergy or ulceration, anorexia, and other relevant symptoms. At the end of the treatment period, blood samples were collected for hematological analysis.

### 2.11. Statistical Analysis

Statistical analysis was performed in GraphPad Prism, version 9 (GraphPad Software Inc., San Diego, CA, USA). Significant differences between the groups were assessed using one-way ANOVA. Comparisons between treatment and control groups were performed using Dunnett’s multiple comparisons test. A value of *p* less than 0.05 was considered statistically significant.

## 3. Results

### 3.1. Effects of Bortezomib and PCC Combination on Lung Cancer Cell Viability 

We tested the cytotoxic effects of bortezomib and PCC in 344SQ, H23, and H358 cancer cell lines after 48 h treatment. Bortezomib alone resulted in a 20–40% reduction in the number of viable cells for these lung cancer cell lines. In addition, PCC treatment as a single treatment had a moderate effect on cell viability, whereas the combination of bortezomib and PCC for 344SQ, H23, and H358 resulted in a 60–90% reduction in the number of viable cells (Figure 1a–c). Further, we tested the effects of PCC on these three cell lines. A significant reduction in the expression of c-FLIP was observed at 250 nM for 344SQ (Figure 1d and Appendix A). However, in H23 and H358, the significant reduction of c-FLIP was observed from 1 µM (Figure 1e,f). These results coincided with the cell viability data. Thus, we combine PCC with bortezomib to achieve maximum effects with less toxicity. 

### 3.2. Role of c-FLIP Overexpression on Cancer Cell Viability

In addition, we sought to confirm whether c-FLIP plays a crucial role in impacting the treatment outcomes. To achieve this, we performed an experiment to assess the effects of bortezomib and PCC combination on the viability of the R331 cell line (clone established from Renca renal cancer cell line) that are overexpressed with c-FLIP [36]. As expected, the combination of bortezomib with PCC caused significant growth inhibition (65.37%) in R331-VC cells when compared with the c-FLIP-overexpressing R331 cells (R331-FLIP) (*p* < 0.0001), (Figure 1e and Appendix A). These results indicated that c-FLIP over expression plays a critical role in maintaining apoptotic resistance of cancer cells.

### 3.3. Bortezomib and PCC Combination on Resting and Activated CD8^+^ T Cells

After we assessed the effects of bortezomib and PCC combination on cancer cells, we sought to see the effects of this combination on CD8^+^ T resting and activated cells. Purified CD8^+^ T cells were activated and treated with bortezomib and PCC combination. Interestingly, the resting T cells showed about 40–50% growth inhibition, but the combination treatment was well tolerated by activated CD8^+^ T cells with no apparent toxicity (Figure 1f). 

### 3.4. Effect of Bortezomib and PCC Combination on Cell Migration 

Further, we explored the effect of bortezomib and PCC on the migration of 344SQ, H23, and H358 cancer cells using a scratch wound healing assay. Our results showed that the migratory capacity of bortezomib and PCC was (25–30%) and (25–35%), respectively. However, the anti-migratory effect of PCC and bortezomib combination was considerably higher (60–80%) when compared with either monotherapy (Figure 2a–c). Further, we confirmed the inhibitory effects on cell migration using Transwell cell migration assay: PCC and bortezomib combination exhibited consistent inhibitory effects on cell migration compared to the untreated cells, indicating a potential anti-migratory effect of PCC + bortezomib (Figure 2e,f).

### 3.5. Impact of Bortezomib and PCC Combination on Invasive Ability of Lung Cancer Cells

Further, we tested the anti-invasive effects of bortezomib and PCC using the cell invasion assay. As shown in Figure 2g–i, the invasion of these lung cancer cells was decreased by 15–40% by bortezomib and 15–35% by PCC, respectively. However, when bortezomib was combined with PCC, the invasive ability of lung cancer cells was decreased to a much higher extent by 60–80%, indicating the potential inhibitory effect on cell invasion by combination treatment compared with either bortezomib or PCC.

### 3.6. Effects of Bortezomib and PCC Combination on c-FLIP Protein Expression

c-FLIP is an important anti-apoptotic protein and a key chemotherapy resistance factor that suppresses chemotherapy-induced apoptosis. Therefore, we sought to analyze the protein expression of c-FLIP in 344SQ, H23, and H358 cells. The protein expression of c-FLIP was higher in 344SQ, H23, and H358 compared with BEAS-2B normal human bronchial epithelium (Figure 3a). Further, we tested the effect of bortezomib and PCC combination on those cells. Hence, bortezomib or PCC as monotherapy showed moderate effects on c-FLIP inhibition, whereas combination treatment of bortezomib with PCC showed more significant inhibition of c-FLIP protein expression in all the three lung cancer cell lines (Figure 3b–d and Appendix A).

### 3.7. Bortezomib and PCC Combination Reduces c-FLIP Protein Stabilization

Upon observing the significant reduction of c-FLIP expression with the combination treatment with bortezomib and PCC, therefore, we sought to analyze the stability of c-FLIP protein in the combination treatment condition in H23 and H358 lung cancer cell lines. As expected, the stability of c-FLIP protein was significantly reduced in combination-treated H23 (t_1/2_ ~7 h; *p* < 0.0001) and H358 (t_1/2_ ~10 h; *p* < 0.001) as compared with the control H23 (t_1/2_ ~16 h) and H358 (t_1/2_ ~16 h) (Figure 3e,f and Appendix A). However, bortezomib or PCC alone decreases the stability of c-FLIP by (t_1/2_ ~15 h; *p* < 0.05) and (t_1/2_ ~14 h; *p* < 0.05), respectively (Appendix A). 

### 3.8. Effects of Bortezomib and PCC Combination on Mitochondrial Respiration of Cancer Cells

Cancer cells are often exposed to a metabolically challenging environments with scarce availability of oxygen and nutrients. Hence, we assessed the effect of bortezomib and PCC co-treatment on cancer cell metabolism using an XFe96 Cell Mito Stress test for mitochondrial oxygen consumption rates (OCR). The basal OCR rate was significantly inhibited in both H23 and H358 cells (Figure 4a–d). Similarly, the maximum respiration was significantly inhibited by combination treatment compared to bortezomib or PCC treatment alone (Figure 4e–g). Strikingly, the ATP production rate was also significantly inhibited by bortezomib and PCC combination in lung cancer (Figure 4g,h) and renal cancer cell lines (Appendix A).

### 3.9. Effects of Bortezomib and PCC Combination on Glycolytic Function on Cancer Cells

It has been reported that cancer cells commonly require an upregulation or metabolic switch towards a glycolytic pathway or to fuel the rapid energy requirements. In addition, cancer cells have the versatility to switch between, or utilize both, glycolysis and mitochondrial respiration through metabolic plasticity. Thus, we performed the Seahorse assay to assess the effect of the bortezomib and PCC combination on the glycolytic function of cancer cells. The results in Figure 5a–d demonstrated that the bortezomib and PCC combination reduced the glycolytic functional parameters, including glycolysis, glycolytic capacity, and glycolytic reserve in H23 cells. Likewise, the combination-treated H358 cells also showed the same trend in inhibiting the glycolysis and the glycolytic reserve (Figure 5a–d).

### 3.10. Antitumor Effects of Bortezomib and PCC Combination in Lung Xenograft Mice

To further evaluate the antitumor efficacy of bortezomib and PCC combination in vivo, first, we injected H358 (human) lung cancer cells subcutaneously in immunocompromised nu/nu NCr mice. Tumor-bearing mice were randomly divided into four groups, as described in the Materials and Methods section, followed by corresponding treatments for four weeks. Tumor-bearing mice treated with bortezomib 1 mg/kg or PCC 10 mg/kg showed 20–40% suppression of tumor growth. Strikingly, mice treated with the combination of bortezomib (1 mg/kg) and PCC (10 mg/kg) showed 60–80% suppression of tumor growth (Figure 6a–c and Appendix A). This tumor growth inhibition was further confirmed by a reduction in tumor weight, which was about 70% in the combination treatment group when compared with monotherapy, where it was 20–30% weight reduction (Figure 6b). Tumor induction and the treatment timeline in the current experiments are as shown in Figure 6d.

Furthermore, the antitumor effect of bortezomib and PCC combination in mice with an intact immune system was determined. The syngeneic 129/Sv mice were inoculated with 344SQ murine lung cancer cell line as indicated in the Materials and Methods section. After the four weeks of treatment period, tumor-bearing mice treated with bortezomib 1 mg/kg or PCC 10 mg/kg showed 25–40% suppression of tumor growth. However, the combination treatment groups showed a significant decrease (60%) in tumor growth (Figure 6e–g and Appendix A). The tumor inhibition was well correlated with the tumor weight (Figure 6f). The working illustration of the bortezomib and PCC combination has been followed as shown in Figure 6h.

### 3.11. In Vivo Toxicity Analysis of Bortezomib and PCC Combination

Treatment of BALB/c mice with bortezomib and PCC combination for 30 days did not cause any adverse effects; there were no observable changes in body weight (Appendix A). Furthermore, no observable changes in food intake, behavior, lethargy, and gastrointestinal toxicity in the combination treatment were observed. Additionally, no abnormal clinical signs or behavior were detected in either of the groups. Hematological observations of all treated mice, including total blood count, red blood cells (RBC), white blood cells (WBC), neutrophils, monocytes, lymphocytes, platelet counts, and hemoglobin levels, were within the normal limits as compared with the control group. No significant differences were noted between the control and treated groups for the hematological parameters measured (Appendix A). However, compared to the control, a significant difference in the WBC was observed only in the bortezomib-treated group, but the PCC and PCC + bortezomib combination groups were comparable with the control group. 

## 4. Discussion

NSCLC accounts for about 85% of all lung cancer cases. Though patients show a promising initial response to therapy, many patients develop resistance, leading to failure in the response to traditional therapeutic options [37]. Despite other causative reasons, overexpression or gene mutations are a significant driver in most lung cancer patients. In addition to the commonly mutated genes, including KRAS, mutation in the p53 gene is present in more than 50% of NSCLC. Thus, it causes resistance to traditional anticancer drugs [15]. Therefore, the main focus of the current study was to maximize the anticancer potential by using low concentrations of bortezomib with PCC and reduce the side effects.

The utilization of bortezomib as an antitumor therapy for various neoplasms has become increasingly significant in recent years. Natural products-based therapeutics has also emerged as a promising approach due to their effectiveness against various cancers with less adverse effects in patients compared to synthetic drugs. Here, we demonstrated that the combination of bortezomib and PCC has a significant antitumor effect on non-small-cell lung cancer (NSCLC) cells. Our results indicated that the loss of viability depends on the concentration of bortezomib and PCC administered in the lung tumor cell lines. Based on our cell viability assay, the optimal response for the 344SQ cell line was observed at 10 nM for bortezomib and 500 nM for PCC. Moreover, the 344SQ cell line was derived from spontaneous tumor of murine background, and the combination of bortezomib and PCC sensitizes the 344SQ cell line at low concentrations when compared to H23 and H358, which are the human-derived cell lines.

Previous studies have reported the ability of bortezomib to decrease gastric cancer viability by reducing pERK levels when combined with other inhibitors and blocking antibodies in gastric cancer models [38]. Additionally, bortezomib and PCC have been found to induce apoptosis in colon cancer cell lines by upregulating TRAIL-induced receptors (TNFRS10 A/B) [39,40], and bortezomib can decrease the growth of cancerous cells by restoring Apaf-1 and increasing the caspase-3 activity [41,42]. Furthermore, bortezomib can synergistically reduce the expression of mutated p53 proteins in NSCLC when used with other treatments that are similar to PCC, such as venetoclax and navitoclax, as mediators of pro-survival proteins [43]. 

Several studies have demonstrated c-FLIP overexpression in various cancers that confers resistance to death-ligand-mediated apoptosis. However, such therapy resistance has been regulated by the downregulation of c-FLIP in cancer cells [44,45]. At the same time, studies reported that a low concentration of PCC was an effective inhibitor of c-FLIP expression in all lung tumors, albeit with varying levels of resistance among cell lines. Notably, our results demonstrated that the inhibition of c-FLIP protein expression was observed at high concentrations in KRAS^mut^, p53^mut^ H23, and H358 cell lines, which contrasts with their basal c-FLIP expression and could be related directly to the loss of cell viability. Studies have demonstrated that c-FLIP can inhibit the proliferation of tumoral cells through the activation of a caspase-dependent mechanism. Results observed in our study are in agreement with an earlier study where the synergic effect of bortezomib in combination with TRAIL inhibitors is reported to reduce the phosphorylation of proliferation pathways such as ½RK 1/2 [38]. 

In our current study, bortezomib by itself was not able to decrease c-FLIP expression in both H23 and H358 cell lines, which might be due to the resistant nature of those cell lines which are mutant for KRAS and p53 [46,47]. A recent study stated that upregulation of the PSMB5 gene in osteosarcoma cell line (~295 nTPM) with an upregulated PSMB5 gene correlation was observed with the increase in the resistance of these cells to the bortezomib treatment [29]. In this context, lung cancer cells with an upregulated PSMB5 H23 (425.1 nTPM (normalized transcript per million)) and H358 (349.9 nTPM) used in our current study are in accordance with the above mechanism, which could be an essential mediator of the effect of bortezomib itself but not in combination with PCC. Interestingly, on the other hand, we have observed that NSCLC cells are more sensitive when bortezomib is combined with PCC. In addition, our results confirmed that the expression of c-FLIP was higher in KRAS^mut^/P53^mut^ condition, and the combination of bortezomib and PCC reduces anti-apoptotic protein c-FLIP by directly targeting bromo and extra terminal domain (BET) proteins and enhancing caspase-8-dependent apoptosis of cancer cells [13]. However, the effect of this combination treatment in KRAS/p53 WT lung cancer cell line is not known. The results of our current study with the combination of bortezomib and PCC are in agreement with a recent study reported by Tewary et al., in which the authors have demonstrated the importance of PCC in sensitizing tumor cells to induce apoptosis through c-FLIP downregulation and increase the effectiveness of other treatments including polyinosinic:polycytidylic acid (poly I:C) [13].

To demonstrate the additional effects of the bortezomib and PCC combination, we sought to evaluate the critical characteristics of cancer cells, including cell migration and invasion. Our findings revealed that, more than individual treatment with bortezomib or PCC, the combination treatment significantly reduced the migration and invasion across all lung cancer cell lines tested in our current study. These results are consistent with a recent study demonstrating bortezomib’s ability to regulate Apaf-1 and its subsequent impact on migration in HER2^+^ breast cancer cell lines [41]. Previous studies reported that the regulation of cell migration and invasion by c-FLIP has been shown to correlate with tumor progression and poor outcomes [48,49]. In addition, the results evidenced that c-FLIP promotes the motility of cancer cells by activating ROCK, FAK, and ERK, and increasing the expression of MMP-9 [48]. Additional evidence was reported demonstrating the role of c-FLIP_L_ in triggering cell motility in ovarian tumors by playing a role in chaperoning tumor cells from immunosurveillance and increasing their invasive potential by augmenting cell motility [50]. Bortezomib and PCC combination was also found to be effective on the mitochondrial respiration of NSCLC by reducing basal and maximal respiration along with a decrease in ATP production in both NSCLC cells. Interestingly, we observed more evident effects in both H23 and H358 cell lines with the higher expression of c-FLIP protein. However, the non-mitochondrial oxygen consumption is a process that takes place outside the mitochondria or at the cell surface and is not impacted by the combination treatment, and thus it stayed at the increased levels. Furthermore, we observed that the combination of bortezomib with PCC reduces the glycolytic function by reducing glycolysis and glycolytic reserve on both cell lines, and thus the combination treatments were found to be more susceptible by effectively decreasing the metabolic reprogramming function of cancer cells [41].

Finally, evaluating the effects of bortezomib and PCC combination in the H358 lung cancer xenograft mice model confirmed a significant decrease in tumor size and weight. Recent studies have shown that PCC can suppress tumor growth in melanoma models when used in combination with other treatments, resulting in a decrease in tumor size and weight [13,51], which is in agreement with our current results. Moreover, administration of bortezomib in combination with other therapies (Everolimus) has been shown to effectively reduce lung metastasis nodules in mice by regulating ERK/MAPK and AKT pathways [52]. Additionally, treatment with bortezomib in combination with CXCR4 inhibitor, a checkpoint inhibitor, has been shown to decrease cholangiocarcinoma tumor mass and volume in mice within 60 days [53]. Notably, bortezomib also enhances immune cell response by regulating p53 proteins. These findings are in support of our current study, where a combination of bortezomib and PCC is a potential therapeutic for treating cancers which are resistant to the currently available classical therapy.

Preliminary hematological analysis of the mice treated with bortezomib or bortezomib as a single agent and in combination with PCC revealed no pathological changes except moderate changes in WBC count in the bortezomib-treated group. However, various anticancer drugs that are already in clinical use have been found to inflict significant toxicities that negatively impact different organs even after single doses of treatment [54]. In addition, >60% of drugs were derived from plants, including anticancer drugs [55]. Therefore, bortezomib in combination with the natural product PCC has the potential to be a safer and more effective anticancer drug against KRAS^mut^/P53^mut^ cancer types [56].

## 5. Conclusions

Overall, the results of the current study demonstrate that PCC sensitizes and significantly improves the anticancer effects of bortezomib on KRAS^mut^/P53^mut^ lung cancer cells, highlighting the potential combination treatments without causing significant side effects.

## Figures and Tables

**Figure 1 cancers-16-00670-f001:**
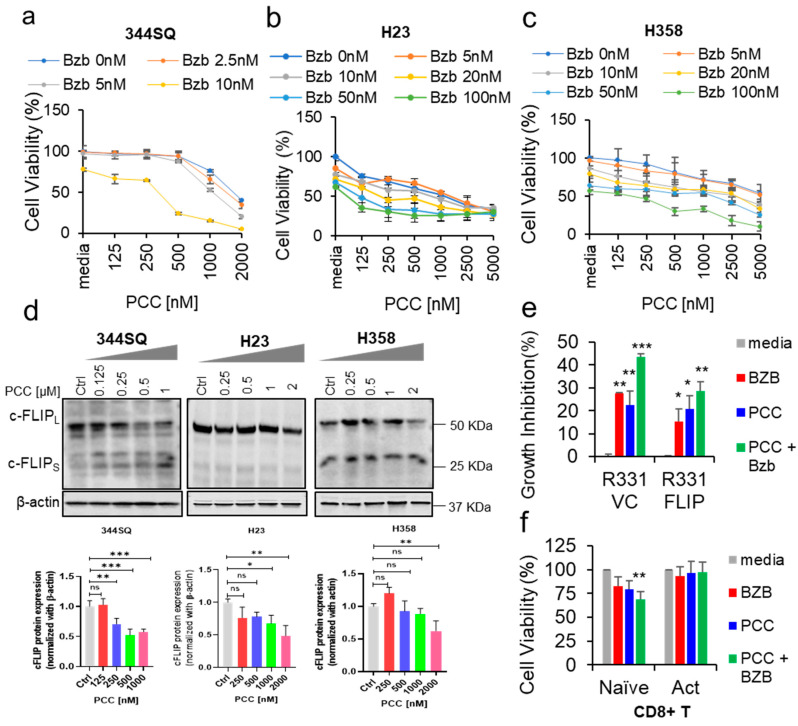
Growth inhibitory effect of bortezomib in combination with PCC on lung cancer cell lines. (**a**–**c**) Cells were treated with indicated concentrations (nM) of PCC for 2–3 h, followed by bortezomib (0–100 nM) for 48 h. Relative cell numbers were assessed by crystal violet staining, followed by reading the plate at 570 nm. (**d**) Western blot analysis of c-FLIP protein expression levels after treatment with different PCC concentrations in 344SQ, H23, and H358 lung cancer cell lines. Bottom panel: Quantification analysis of c-FLIP protein levels in the respective cell lines. Original Western Blot images can be found in Appendix A. (**e**) Cell viability assay was performed in c-FLIP overexpressed and vector control renal cells after treatment with bortezomib and PCC with indicated concentrations. (**f**) Cells from the spleen and lymph node were isolated from BALB/c mice, purified using a CD8^+^ T cell purification kit containing an antibody cocktail, and activated with anti-CD3 and anti-CD28 for 48 h with the presence of bortezomib and PCC. Then, the cell viability was assessed using MTS assay following the kit protocol. * *p* < 0.05, ** *p* < 0.01, *** *p* < 0.001, n.s. not significant.

**Figure 2 cancers-16-00670-f002:**
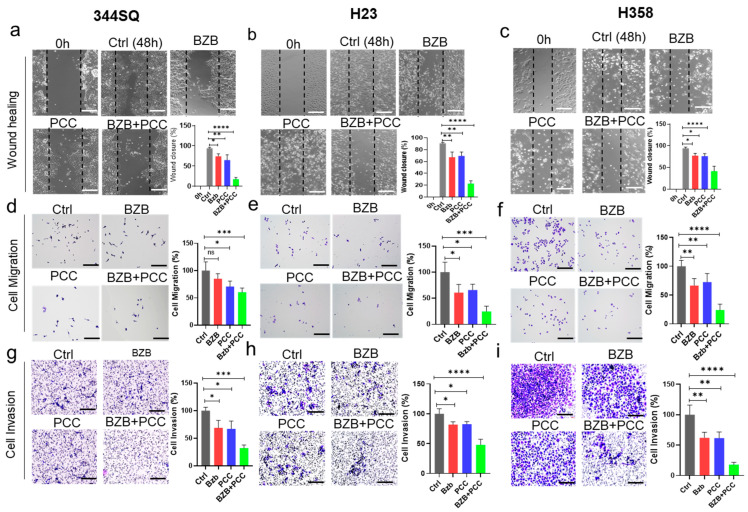
Effects of bortezomib and PCC combination on migration and invasion of lung carcinoma cells. (**a**–**c**) 344SQ, H23, and H358 cells were grown, and the scratches were made, followed by treating the cells with indicated concentrations of bortezomib, PCC, and combination 344SQ-BZB 5 nM/PCC 250 nM; H23 and H358-BZB 20 nM/PCC-500 nM for 48 h and the wound closure was quantified using ImageJ (Version 1.53e). The bar graph shows the quantification analysis of cell migration compared with the untreated controls. (**d**–**f**) Transwell cell migration and (**g**–**i**) invasion assays were performed after treating with bortezomib and PCC as indicated above (refer to the Materials and Methodology section for detailed protocol). Scale bar is 100 µm. * *p* < 0.05, ** *p* < 0.01, *** *p* < 0.001, **** *p* < 0.0001, n.s. not significant.

**Figure 3 cancers-16-00670-f003:**
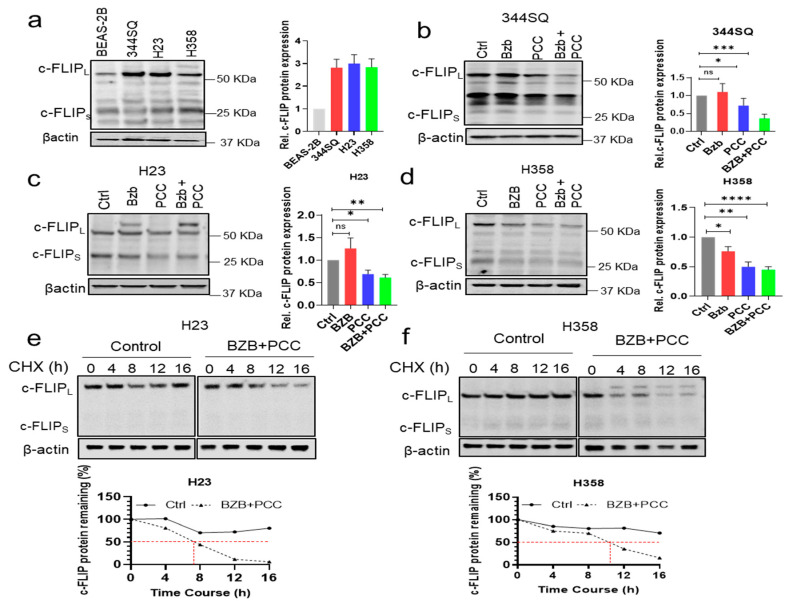
Effects of bortezomib and PCC combination treatment on c-FLIP expression in lung cancer cells. (**a**) Western blot showing the protein expression levels of c-FLIP in BEAS-2B (human non-tumorigenic lung epithelial cell line), 344SQ (murine lung adenocarcinoma cells with high metastatic potential), H23, and H358 (human lung adenocarcinoma, NSCLC). Graph showing the quantification analysis of c-FLIP protein levels in the respective cell lines. (**b**–**d**) combinatorial effects of bortezomib and PCC in (**b**) 344SQ, (**c**) H23, and (**d**) H358 cell lines. (**e**,**f**) The effects of bortezomib and PCC combination treatment on c-FLIP protein stabilization performed using Cycloheximide (CHX) chase assay. Briefly, H23 and H358 cells were treated with CHX (100 µg/mL) at indicated time points, and cell lysates were prepared followed by Western blotting analysis. Bottom panel: Quantification analysis of c-FLIP protein stabilization in the respective cell lines and the red dotted line shows the t_1/2_ of the protein * *p* < 0.05, ** *p* < 0.01, *** *p* < 0.001, **** *p* < 0.0001, n.s. not significant. Original Western Blot images can be found in Appendix A.

**Figure 4 cancers-16-00670-f004:**
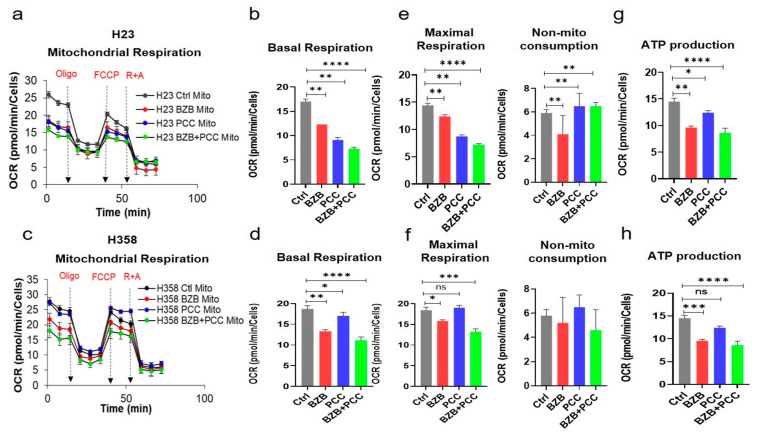
Modulatory effects of bortezomib and PCC combination on mitochondrial respiration of lung cancer cells. (**a**) Mitostress assay was performed by assessing oxygen consumption rates (OCR) profile of monolayer cells using an XF96 Seahorse analyzer after 48 h treatment with the indicated concentration of bortezomib and PCC combination in the presence of oligomycin (3 μM), FCCP (0.25 μM), and rotenone (1 μM) + antimycin (1 μM). (**b**) Mitostress assay bioenergetic parameters measured from Seahorse results: (**b**,**d**) basal respiration, (**e**,**f**) maximal respiration, and (**g**,**h**) ATP production were also assessed. (**c**,**d**) Mitochondrial respiratory parameters on H358 lung cancer cell line were performed as indicated above (**c**–**h**). OCR values were normalized on protein content measured by SRB assay as per the manufacturer’s instructions. * *p* < 0.05, ** *p* < 0.01, *** *p* < 0.001, **** *p* < 0.0001, n.s. not significant.

**Figure 5 cancers-16-00670-f005:**
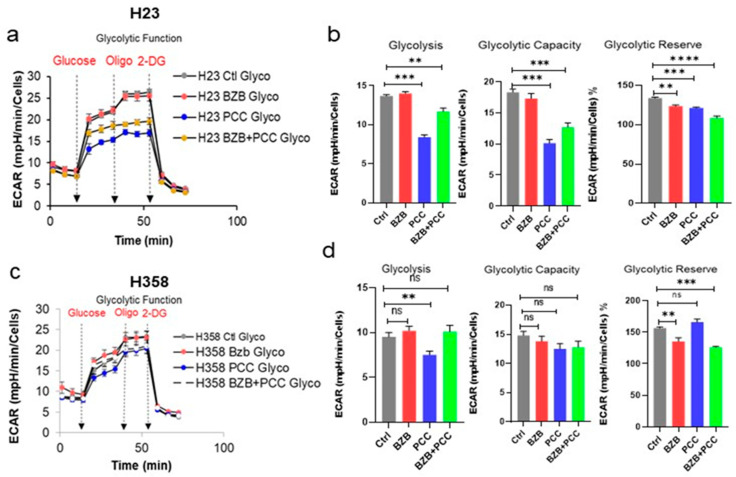
Effects of bortezomib and PCC combination on glycolytic function of lung cancer cells. (**a**) Representative Extra Cellular Acidification Rate (ECAR) profile of monolayer cells subjected to XF Glycolysis Stress Test with an XFe96 Agilent Seahorse under sequential injections of 10 mM of glucose, 3 µM of oligomycin A, and 100 mM of 2-deoxy-D-glucose. (**b**) Glycolytic assay bioenergetic parameters measured from Seahorse results: glycolysis, glycolytic capacity, and glycolylic reserve were assessed. ECAR values were normalized on protein content measured by SRB assay as per the manufacturer’s instructions. (**c**,**d**). ECAR profiles and the glycolytic bioenergetic parameters of H358 cell line after bortezomib and PCC treatment as indicated above. Statistical test: *t*-test, ** *p* < 0.01, *** *p* < 0.001, **** *p* < 0.0001, n.s. not significant.

**Figure 6 cancers-16-00670-f006:**
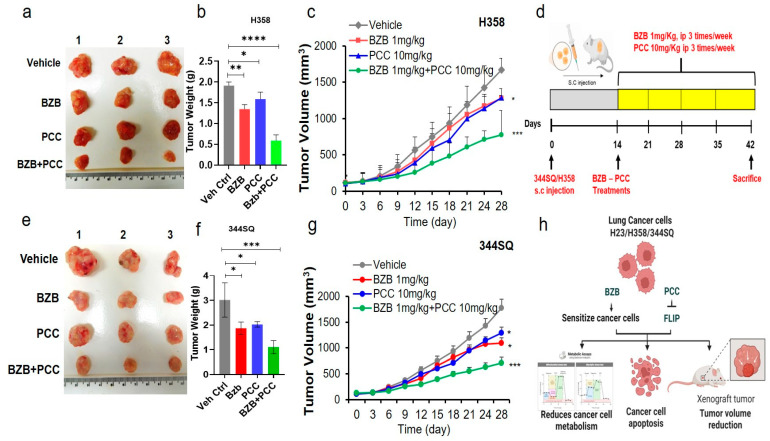
Antitumorigenic effects of bortezomib and PCC combination on H23 lung tumor xenograft mouse model. (**a**) H23 lung cancer cells were injected into the right flank of NSG mice on day 0. On day 14, when the xenografted tumors were established, mice were randomized into four groups (n = 6 in each group). Group I was the untreated control; group II was treated with bortezomib (1 mg/kg; 3 times/week; i.p. injection); group III was treated with PCC (10 mg/kg, 3 times/week; s.c. injection); and group IV was treated with the combination of bortezomib (1 mg/kg; 3 times/week; i.p. injection) and PCC 10 mg/kg, 3 times/week; s.c. injection) for 4 weeks. (**b**) Tumor weight of the mice with and without treatment. (**c**) Tumor volume was measured by caliper twice every week. All the experimental mice were sacrificed, and tumors were excised and weighed. Volume of excised tumors after four weeks of treatment. (**d**) Tumor establishment and treatment timeline for bortezomib and PCC combination treatment. (**e**) Antitumorigenic effects of bortezomib and PCC combination on 129/Sv syngeneic mouse model. (**f**) Tumor weight. (**g**) Tumor volume measurement over the treatment period between vehicle control and treatment groups. (**h**) Illustration of bortezomib and PCC combination on lung cancer cell lines and tumor xenograft mouse model (* *p* < 0.05, ** *p* < 0.01, *** *p* < 0.001, **** *p* < 0.0001).

## Data Availability

The original contributions presented in this study are included in this article and its Appendix A, further inquiries can be directed to the corresponding author.

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
