# Peer review of "Bortezomib in Combination with Physachenolide C Reduces the Tumorigenic Properties of KRASmut/P53mut Lung Cancer Cells by Inhibiting c-FLIP"

_cancers, 2024, doi:10.3390/cancers16030670_

Round 1
Reviewer 1 Report
Comments and Suggestions for Authors
The authors in this study have studied the effect of effects of a natural product, physachenolide C (PCC) and bortezomib, in KRASmut/P53mut lung cancer cells The authors claim that PCC sensitizes cancer cells to bortezomib, potentially improving the antitumor effects against KRASmut/P53mut lung cancer cells, with an enhanced efficacy of combination treatments. The study is good but the manuscript can be made comprehensive by working on the following points.
1. The English language and scientific needs a check.
2. In Figure1-a, why the authors have used limited concentrations in 344SQ as compared to the Fig.1-b&C.
3. The B-Actin blot in the Fig.1-d(H358) and Fig.3-a can be better.
4. Figure 2 needs better resolution and better labelling/legends of Fig.2, h & i and Fig.3
5. Figure 4,-b what is the reason for high non-mito consumption, is it as per the experiment?
6. Figure. 6 needs better resolution and labelling.
Comments on the Quality of English Language
The English language and scientific needs a check.
Author Response
We would like to thank the reviewer for taking the time to review this manuscript. Please find the detailed responses below and the corresponding revisions/corrections highlighted/in track changes in the re-submitted files.
Point by point responses to Reviewers’ comments on Manuscript ID: cancers -2759800
We would like to thank the reviewers for recognizing the significance of our findings. The comments from reviewers are very positive and constructive. We have addressed the reviewers’ comments point by point in full by including the required data by performing additional experiments and by providing the explanations in the revised version. We believe that our revised manuscript has been strengthened tremendously and now is suitable for publication in the journal Cancers.
Reviewer 1: The authors in this study have studied the effect of effects of a natural product, physachenolide C (PCC) and bortezomib, in KRASmut/P53mut lung cancer cells. The authors claim that PCC sensitizes cancer cells to bortezomib, potentially improving the antitumor effects against KRASmut/P53mut lung cancer cells, with an enhanced efficacy of combination treatments. The study is good but the manuscript can be made comprehensive by working on the following points.
1. The English language and scientific needs a check.
Response: The English language and the Scientific contents were checked and corrected.
2. In Figure1-a, why the authors have used limited concentrations in 344SQ as compared to the Fig.1-b&C.
Response: We thank the reviewer for the critical question. In our initial experiments, we used the same range of concentrations for bortezomib (0-100nM) and PCC (0- 5µM) in 344SQ cell line. Based on the cell viability assay, the optimal response was observed at 10nM for bortezomib and 500nM for PCC. Moreover, 344SQ cell line was derived from spontaneous tumor of murine background and the combination of bortezomib and PCC sensitizes 344SQ cell line at low concentrations when compared with H23 and H358 shown in figure b and c. Which has been indicated in the discussion section.
3. The B-Actin blot in the Fig.1-d(H358) and Fig.3-a can be better.
Response: As suggested, the β-actin band has been replaced with low exposure blots for Fig. 1d and Fig. 3a.
4. Figure 2 needs better resolution and better labelling/legends of Fig.2, h & i and Fig.3
Response: As suggested by the reviewer, the resolution and the labelling has been corrected in Fig. 2h and i and Fig.3
5. Figure 4,-b what is the reason for high non-mito consumption, is it as per the experiment?
Response: We appreciate the reviewer’s question. Non-mitochondrial oxygen consumption is a process that takes place outside the mitochondria or at the cell surface. This type of non-mitochondrial oxygen consumption process is not impacted by the combination treatment and thus it stayed at the increased levels. However, the combination of bortezomib and PCC inhibit the mitochondrial respiration by reducing mitochondrial maximal respiration and ATP production (Fig 4a). It is explained in the discussion section.
6. Figure. 6 needs better resolution and labelling.
Response: A per the reviewer’s suggestion, the resolution of Fig.6 has been enhanced and better labelled.

Reviewer 2 Report
Comments and Suggestions for Authors
The author demonstrated that PCC exerts the anti-cancer effect of bortezomib both in vitro and in vivo, particularly with KRASmut/P53mut lung cancer cells. This work is interesting, however, there are many points required clarifications.
1. Does PCC effect particular on anti-apoptotic FLIP? How was another anti-apoptotic proteins? Why does PCC have specificity to FLIP?
2. Result section 3.2, the author mention that Renca-FLIP is FLIP overexpressing cells. However, there is no experiment confirming FLIP expression.
3. Fig 2, what is the concentration for PCC and bortezomib used in cell migration and invasion assay? Please include in the Figure legend. These concentrations have toxic to the cells? If yes, the inhibitory effect of PCC and bortezomib might be a result of cell death induction, not the actual effect on cell migration and invasion. For wound healing assay, the data at 0 h of all treatments should be included. The font size in the plot is too small.
4. Fig 2D and E, the control group has less cell migration, therefore the inhibitory effect of the treatment is not convincing. And the image of invasion assay was not clear, so that I cannot assess.
5. How FLIP regulate cell migration and invasion?
6. Fig 3, what is the concentration of the treatments? The reduction of FLIP in response to PCC was not consistent. As compared to Fig 1D, PCC has greater effect on FLIP reduction, but in Fig 3B, PCC has no significant effect on FLIP level. Fig 3C-D, the author conclude that combination treatment has more significant on FLIP reduction as compared to single treatment. However, the blots were not convincing.
7. Fig 3E and F, how was the single treatment effect FLIP stability? If the author conclude that combination treatment has more potent effect, FLIP stability in response to treatment with PCC or bortezomib alone should be included.
8. The author concluded that “PCC sensitizes and significantly improves the anticancer effects of bortezomib on KRASmut/P53mut lung cancer cells”. Does this combination also affect in wt lung cancer cells? Or they have specific only KRASmut/P53mut lung cancer cells? Why they have specific to mutant cells?
9. The approval number for in vivo experiment should be included.
10. The figures were not appropriated for scientific presentation. Please adjust the font size for readability.
Comments on the Quality of English Language
N/A
Author Response
Point by point responses to Reviewers’ comments on Manuscript ID: cancers -2759800
We would like to thank the reviewers for recognizing the significance of our findings. The comments from reviewers are very positive and constructive. We have addressed the reviewers’ comments point by point in full by including the required data by performing additional experiments and by providing the explanations in the revised version. We believe that our revised manuscript has been strengthened tremendously and now is suitable for publication in the journal Cancers.
Reviewer 2: The author demonstrated that PCC exerts the anti-cancer effect of bortezomib both in vitro and in vivo, particularly with KRASmut/P53mut lung cancer cells. This work is interesting, however, there are many points required clarifications.
1. Does PCC effect particular on anti-apoptotic FLIP? How was another anti-apoptotic proteins? Why does PCC have specificity to FLIP?
Response: We appreciate the reviewer’s valuable comments. Results from our earlier study (Tewary et al., Cancer Research 2021) confirmed that PCC inhibit cFLIP by specifically targeting BET proteins to reduce antiapoptotic proteins and enhance caspase-8–dependent apoptosis of cancer cells.
2. Result section 3.2, the author mention that Renca-FLIP is FLIP overexpressing cells. However, there is no experiment confirming FLIP expression.
Response: We thank the reviewer for pointing out this question. The cFLIP overexpressing cell line was generated as described in our earlier studies and had been cited in the methods section (Naoko Seki et al., Cancer Research 2003 and Tewary et al., Cancer Research 2021). In this study, the cFLIP overexpressed cell line has been utilized to confirm its critical role in resistance mechanism against treatment strategies. The detailed description about the cell lines have been included in the methodology section.
3. Fig 2, what is the concentration for PCC and bortezomib used in cell migration and invasion assay? Please include in the Figure legend. These concentrations have toxic to the cells? If yes, the inhibitory effect of PCC and bortezomib might be a result of cell death induction, not the actual effect on cell migration and invasion. For wound healing assay, the data at 0 h of all treatments should be included. The font size in the plot is too small.
Response: We thank the reviewer for the comments and questions. We used the following treatment concentrations: for 344SQ-BZB 5nM/PCC 250nM; H23 and H358-BZB 20nM/PCC-500nM. In addition, as per the data in figure 1 a-c, the concentration we chose did not cause significant toxicity to the cells. Moreover, representative images of 0h treatment (showing same wound gaps) of the scratch assay are included. Also, the font size in the plots in figure 2 has been corrected.
4. Fig 2D and E, the control group has less cell migration, therefore the inhibitory effect of the treatment is not convincing. And the image of invasion assay was not clear, so that I cannot assess.
Response: We thank the reviewer for pointing this comment. We enhanced the clarity of image and the same has been replaced in figure 2d and e.
5. How FLIP regulate cell migration and invasion?
Response: A valid question, which has been addressed by incorporating the following justification in the discussion part in the current manuscript. Previous studies reported that the regulation of cell migration and invasion by c-FLIP has been shown to correlate with tumor progression and poor outcomes (Park et al., 2008, Shim, E et al., 2007). In addition, it is also evidenced that c-FLIP promotes the motility of cancer cells by activating ROCK, FAK, and ERK, and increasing the expression of MMP-9 (Park et al., 2008). Additional evidence demonstrating the role of c-FLIPL in triggering cell motility in ovarian tumors by playing a role in chaperoning tumor cells from immunosurveillance and increasing their invasive potential by augmenting cell motility (El-Gazzar, A et al., 2010).
6. Fig 3, what is the concentration of the treatments? The reduction of FLIP in response to PCC was not consistent. As compared to Fig 1D, PCC has greater effect on FLIP reduction, but in Fig 3B, PCC has no significant effect on FLIP level. Fig 3C-D, the author conclude that combination treatment has more significant on FLIP reduction as compared to single treatment. However, the blots were not convincing.
Response: We appreciate the reviewer’s comments and questions. In figure 1d, we assessed the dose dependent effects of PCC ranging from 0-1µM for 344SQ. However, in figure 3b, we chose to treat the cells with 250nM concentration of PCC and the expression of c-FLIP was consistent with figure 1d. In addition, the quantification of band intensity of the C-FILP inhibition in figure 3c and d showed a significant difference in the combination group compared with either bortezomib or PCC treatment alone.
7. Fig 3E and F, how was the single treatment effect FLIP stability? If the author conclude that combination treatment has more potent effect, FLIP stability in response to treatment with PCC or bortezomib alone should be included.
Response: We appreciate the reviewer for pointing this question. As suggested, the FLIP stability for PCC and bortezomib has been included in the supplementary figure 1. The results indicated that bortezomib or PCC would decrease the stability of c-FLIP by 15 and 30 % respectively. However, the combination treatment decreases the stability of c-FLIP by 90%.
8. The author concluded that “PCC sensitizes and significantly improves the anticancer effects of bortezomib on KRASmut/P53mut lung cancer cells”. Does this combination also affect in wt lung cancer cells? Or they have specific only KRASmut/P53mut lung cancer cells? Why they have specific to mutant cells?
Response: We appreciate the reviewer for pointing this important question. Our results confirmed that the expression of c-FLIP was higher in KRASmut/P53mut condition. In addition, recent studies established bromo and extra terminal domain (BET) proteins as major cellular targets of PCC (Naoko Seki et al., Cancer Research 2003; Tewary et al., Cancer Research 2021). Thus, by targeting of BET proteins to reduce the downstream anti-apoptotic protein cFLIP and enhance caspase 8-dependent apoptosis of cancer cells. However, the focus of this manuscript is on KRASmut/P53mut lung cancer types which promote tumor growth and metastasis as compared to WT lung cancer cells.
9. The approval number for in vivo experiment should be included.
Response: As suggested, the approval number for the animal experiments has been included in the method section.
10. The figures were not appropriated for scientific presentation. Please adjust the font size for readability.
Response: We appreciate the reviewer’s comments. The font size has been corrected for all the figures.
List of References:
- Tewary P, Brooks AD, Xu YM, Wijeratne EMK, Babyak AL, Back TC, Chari R, Evans CN, Henrich CJ, Meyer TJ, Edmondson EF, de Aquino MTP, Kanagasabai T, Shanker A, Gunatilaka AAL, Sayers TJ. Small-Molecule Natural Product Physachenolide C Potentiates Immunotherapy Efficacy by Targeting BET Proteins. Cancer Res. 2021 Jun 15;81(12):3374-3386. doi: 10.1158/0008-5472.CAN-20-2634. Epub 2021 Apr 9. PMID: 33837043; PMCID: PMC8802328.
- Seki N, Hayakawa Y, Brooks AD, Wine J, Wiltrout RH, Yagita H, et al. Tumor necrosis factor‐related apoptosis‐inducing ligand‐mediated apoptosis is an important endogenous mechanism for resistance of liver metastases in murine renal cancer. Cancer Res 2003;63:207‐213.
- , P.; Anees, M.; Horvat, R.; Mikulits, W.; Grunt, T.W.; Mayer, B.; Krainer, M. The role of c-FLIP in ovarian cancer: Chaperoning tumor cells from immunosurveillance and increasing their invasive potential. Oncol. 2010, 117, 451-459.
- Shim, E.; Lee, Y.S.; Kim, H.Y.; Jeoung, D. Down-regulation of c-FLIP increases reactive oxygen species, induces phosphorylation of serine/threonine kinase Akt, and impairs motility of cancer cells. Lett. 2007, 29, 141-147.
- Park, D.; Shim, E.; Kim, Y.; Kim, Y.M.; Lee, H.; Choe, J.; Kang, D.; Lee, Y.S.; Jeoung, D. c-FLIP promotes the motility of cancer cells by activating FAK and ERK, and increasing MMP-9 expression. Cells 2008, 25, 184-195.

Round 2
Reviewer 2 Report
Comments and Suggestions for Authors
The authors have clarified some of my concerns, however, most of them are remaining in the revised manuscript.
1. As my suggestion in comment#2, the additional blot of FLIP expression in FLIP overexpressing cells as compared to the control cells would make the MS more scientifically complete.
2. Also in comment#3, the concentration of treatment should be included in Fig legend.
3. In comment#4, the author mentioned about the migration assay in line 165-166 that " the cell migratory effect was calculated as the percentage of the cell migration." But Fig 2D-F, the plots show number of migrated cells. In addtion, the number of migrated cells in the image are not high to 100 cells particulary in control group, and also for data analysis in invasion assay. The scale bar should be also included.
4. In comment#6, as the authors replied "In figure 1d, we assessed the dose dependent effects of PCC ranging from 0-1µM for 344SQ. However, in figure 3b, we chose to treat the cells with 250nM concentration of PCC and the expression of c-FLIP was consistent with figure 1d." This is not convining. Fig 1D, PCC 250 nm show a significnat reduction with appeoximately 70% protein left, while PCC 250 nm show no significant alteration in Fig 3B.
5. In comments#7, the author should include the value of T1/2 and significant analysis.
6. In comments#8, the author should discussed this point in discussion part.
Comments on the Quality of English Language
N/A
Author Response
Point by point responses to Reviewers’ comments on Manuscript ID: cancers -2759800
We would like to thank the reviewers for recognizing the significance of our findings. We have addressed the reviewers’ comments point by point in full by including the required data by performing additional experiments and by providing the explanations in the revised version. We believe that our revised manuscript has been strengthened tremendously and now is suitable for publication in the journal Cancers.
1. As my suggestion in comment#2, the additional blot of FLIP expression in FLIP overexpressing cells as compared to the control cells would make the MS more scientifically complete.
Response: We appreciate the reviewer’s suggestion. The Western blot for cFLIP overexpressing cells has been performed and the image has been included in the supplementary figure 1.
2. Also in comment#3, the concentration of treatment should be included in the figure legend.
Response: As suggested, the concentration of the treatment has been included in figure 2 legend.
3. In comment#4, the author mentioned about the migration assay in line 165-166 that " the cell migratory effect was calculated as the percentage of the cell migration." But Fig 2D-F, the plots show number of migrated cells. In addition, the number of migrated cells in the image are not high to 100 cells particularly in control group, and also for data analysis in invasion assay. The scale bar should be also included.
Response: We thank the reviewer for pointing out this question and the comments. For the migration and invasion assay, the number of migrated and invaded cells were counted in four random microscopic fields and the representative images have been shown. In addition, now we calculated the number of migrated cells as percentage and the same has been plotted in figure 2. The scale bar has been included.
4. In comment#6, as the authors replied "In figure 1d, we assessed the dose dependent effects of PCC ranging from 0-1µM for 344SQ. However, in figure 3b, we chose to treat the cells with 250nM concentration of PCC and the expression of c-FLIP was consistent with figure 1d." This is not convincing. Fig 1D, PCC 250 nm show a significant reduction with approximately 70% protein left, while PCC 250 nm show no significant alteration in Fig 3B.
Response: We thank the reviewer for pointing out this question. The new Western blot image has been performed and the image with similar exposure to fig 1d has been included. Now the quantification of cFLIP expression is almost similar with figure 3b and the graph has been included.
5. In comments#7, the author should include the value of T1/2 and significant analysis.
Response: As suggested, the t1/2 value and the significance analysis has been included in the results section in figure 3.
6. In comments#8, the author should discussed this point in discussion part.
Response: We included those points in the discussion as suggested.
Round 3
Reviewer 2 Report
Comments and Suggestions for Authors
The authors have clarified most of my concerns.
Comments on the Quality of English Language
Some minor errors were found.